# Mechanical Properties of Thermoplastic Composites Made of Commingled Carbon Fiber/Nylon Fiber

**DOI:** 10.3390/polym13193206

**Published:** 2021-09-22

**Authors:** Mizuki Ono, Masachika Yamane, Shuichi Tanoue, Hideyuki Uematsu, Yoshihiro Yamashita

**Affiliations:** 1Frontier Fiber Technology and Science, University of Fukui, 3-9-1 Bunkyo, Fukui 910-8507, Japan; mizuki_55@outlook.jp (M.O.); tanoue@u-fukui.ac.jp (S.T.); uematsu@matse.u-fukui.ac.jp (H.U.); 2Headquarters for Innovative Society-Academia Cooperation, University of Fukui, 3-9-1 Bunkyo, Fukui 910-8507, Japan; myamane@u-fukui.ac.jp; 3Research Center for Fibers and Materials, University of Fukui, 3-9-1 Bunkyo, Fukui 910-8507, Japan

**Keywords:** commingle yarn, carbon fiber, spread yarn fabric, nylon fiber, composites

## Abstract

Commingled yarns consisting of thermoplastic nylon fibers and carbon fibers can be used to produce superior carbon fiber reinforced thermoplastics (CFRTP) by applying fiber spreading technology after commingling. In this study, we examined whether spread commingled carbon fiber/nylon fiber yarns could reduce the impregnation distance, as there are few reports on this. From this study, the following are revealed. The impregnation speed of the nylon resin on the carbon fiber was very fast, less than 1 min. As the molding time increased, the tensile strength and tensile fracture strain slightly decreased, and the nylon resin deteriorated. The effects of molding time on flexural strength, flexural modulus, and flexural fracture strain were negligible. From the cross-sectional observation conducted to confirm the impregnation state of the matrix resin, no voids were observed in the molded products, regardless of molding time or molding pressure, indicating that resin impregnation into the carbon fiber bundle of the spread commingled yarn fabric was completed at a molding pressure of 5 MPa and a molding time of 5 min.

## 1. Introduction

With the introduction of global CO_2_ regulations and rising fuel prices, the need for lighter cars has dramatically increased. Although there are options for reducing the weight of automobiles, such as the use of high-tensile steel plates and large quantities of aluminum alloy, the application of carbon fiber reinforced plastics (CFRP), which have excellent specific strength and modulus, to structural elements is considered the most efficient. CFRP is a composite material that uses plastic as a matrix material and carbon fiber as a reinforcement material. It should be used in a wide range of fields due to its characteristics of high strength, high elastic modulus, and light weight (low density). Depending on the matrix resin used as the basic material, CFRP can be divided into two types: thermosetting CFRP, which uses epoxy or unsaturated polyester, and thermoplastic CFRP which uses polypropylene or nylon. Thermosetting CFRP is a flexible material prior to hardening due to the nature of the thermosetting resin, making it easy for the resin to impregnate the carbon fiber bundle and relatively easy to fabricate the prepreg sheet, which is the intermediate material for CFRP. However, thermosetting resins need more curing time because they undergo crosslinking reactions by chemical reactions.

On the other hand, thermoplastics do not require a thermosetting process and can reversibly repeat the solid state at room temperature and the molten state at high temperatures without any chemical reaction, which means that molding can be carried out in a very short time. Therefore, it requires only the cooling of the material and leads to a short molding cycle. In addition, it can be remolded by reheating and has excellent recyclability. Therefore, if it is possible to provide thermoplastic CFRP with, for example, higher strength and a higher elastic modulus, comparable to those of thermosetting CFRP, this will be expected to serve as a material with high productivity and excellent recyclability.

Demand for continuous fiber reinforced thermoplastic composites, which have continuous fibers, is expected to increase significantly, as the strength of the reinforcing fibers can be maximized for use in the structural materials of automobiles and aircraft. However, the melting viscosity of the thermoplastic resin is extremely high compared to that of the pre-hardening thermosetting resin. Therefore, the challenge is impregnating the reinforcing fiber bundle with thermoplastic resin in a short time. Preimpregnated thermoplastic resin into carbon fiber bundle (thermoplastic prepreg) is difficult to adapt to molded products with complex shapes unless the viscosity of the resin is reduced by preheating. The relationship between the resin/fiber distance and the resin penetration time was compared with the analytical model using the Darcy rule and the experimental results [1,2,3,4,5,6]. Several molding methods have been proposed for the manufacture of prepreg thermoplastic resin and carbon fiber, including the powder method [7,8,9], the commingled fiber method [10,11,12,13,14,15,16,17,18,19], and the film-stacking method [20,21,22,23,24,25], and the relationships between these molding methods and the degree of impregnation have been reported. The results show that the thermoplastic prepreg with the best impregnation properties is the one made of commingled yarn, which has the shortest distance between carbon and thermoplastic fibers. Consequently, it is necessary to establish a molding technology to produce CFRTP using commingled yarn.

On the other hand, to fully demonstrate the strength of carbon fiber, one study shortened the impregnation distance in the film stacking method by applying “open continuously reinforcing fiber tow,” a technique to thinly and widely spread carbon fiber bundles [26]. It is also known that the “thin-ply effect” can be obtained, which improves the mechanical properties of the composite materials by reducing the thickness of the prepreg to an extremely low level [27,28]. It is expected that a further fiber-spreading treatment of commingled yarns, consisting of thermoplastic and carbon fibers, could produce superior CFRTP, but few studies have been conducted toward that end. In this study, we fabricated woven fabrics using spread commingled yarns, and investigated the effects of forming time and pressure in press forming on the mechanical properties of the molded products.

## 2. Materials and Methods

### 2.1. Material

Commingled yarn consisting of polyacrylonitrile (PAN)-based carbon fiber (CF) and low-water-absorbing nylon resin fiber was used as the material for the woven thermoplastic CFRP laminate. Table 1 shows the specifications of the commingled yarn. First, the yarns were spread to a width of 30 mm, and then the yarns were coated with 15 wt% copolymerized polyamide powders to seal the spread commingled yarns. These were then woven with a rapier loom. Table 2 shows the properties of the spread commingled yarn fabrics. Figure 1 shows the procedure for spread and bonding commingled yarns. Figure 2 shows a photograph of the spread plain fabric using commingled yarn. This fabric was cut into 245 mm squares, and these fabrics were point-bonded to each other using a soldering iron with a temperature-control function (RX-802AS manufactured by Taiyo Denki Sangyo, Japan), then individually laminated (Figure 3). The temperature of the soldering iron was set to 270 °C. The number of layers was set to 30 so that the thickness of the molded product would be approximately 2 mm. Table 3 shows the specifications of the laminated fabrics made from spread commingled yarn.

### 2.2. Press Molding

A heating and cooling press-molding machine (UFHS2500, H&F Co. Ltd., Fukui, Japan) was used as a press-molding machine. Figure 4 shows a photograph of the machine. The heating plate and the cooling plate were lined up next to each other, and a die lifter was installed in front of and behind of the two plates. As a result, heated molds were transported on a cooling plate using a die lifter for quick cooling. A male—female mating system was adopted for molding. The mold area dimensions were 250 mm square.

### 2.3. Press-Forming Conditions

The press-forming process used in this experiment is described below. The molding temperature and pressure were based on the study by Uematsu et al. [23].
The spread commingled yarn fabric was laminated in advance using any number of layers and any lamination configuration, then dried in a vacuum dryer for at least 15 h at 80 °C.After it was confirmed that the mold had reached the desired molding temperature, the laminated spread commingled yarn fabric was removed from the vacuum dryer and fed into the mold.To melt the nylon fibers, preheating was performed under 1 MPa pressure for 5 min.The mold was maintained at a predetermined pressure (molding pressure) and for a predetermined time (molding time).After the heat molding was completed, the mold was unclamped, lifted off the hot plate by the die lifter, transferred from the heating plate to the cooling plate at room temperature, and cooled again by applying the desired pressure (molding pressure).After the mold temperature was confirmed to be below 80 °C the mold was opened by the press machine and the molded product was demolded using ejector plate.

To study the effects of molding pressure on the mechanical properties of the molded products, molding was carried out at two pressure levels: 5 MPa and 10 MPa. The molding temperature was set to 270 °C and the molding time was set to four levels: 5, 10, 20, and 30 min (Table 4). Figure 5 shows an example of the temperature and pressure history during the molding, where the molding pressure is 5 MPa and the molding time is 30 min.

### 2.4. Tensile Test and Four-Point Flexural Test

The tensile test was carried out as described in JIS K 7164. A CCD camera was used to measure the increase in the distance between the sample reference lines during the test. The four-point flexural test was then performed according to JIS K 7074. Flexural deflection was measured with a flexural deflectometer placed directly under the center of the test specimen. Table 5 and Table 6 present the test conditions of the tensile test, and the four-point flexural test, respectively. For both tests, the fabric was cut so that the warp direction of the fabric was in the longitudinal direction of the specimen. A precision universal testing machine (AG-IS100kN, Shimadzu Co. Ltd., Kyoto, Japan) was used for the tensile test. Figure 6 shows a photograph of the four-point flexural test’s jig.

### 2.5. Measurement of Fiber Volume Content

To measure the volume content of carbon fiber *V_f_*, the combustion method was adopted, and the matrix resin was completely burned in an electric furnace at a combustion temperature of 400 °C for more than 15 h. The densities of the carbon fiber and nylon were *ρ_f_* = 1.82 × 10^3^ kg/m^3^ and *ρ_m_* = 1.11 × 10^3^ kg/m^3^, respectively.

### 2.6. Confirmation of Resin Impregnation to the Interior of the Carbon Fiber

To observe the condition of the impregnated resin in the carbon fiber bundle, a cross-section of the molded product was observed. The center of the flat molded plate was cut into a 15 mm square using a composite cutting machine, and the cut sample was encapsulated in unsaturated polyester resin. The cross-section with a mirror finish was observed using a microscope (VHX-2000, Keyence Co. Ltd., Osaka, Japan).

## 3. Results

### 3.1. Fiber Volume Content

The fiber volume fraction *V_f_* of each molded product measured by the combustion method is shown in Table 7. The measured results of *V_f_* were 38.3%~38.9% for the products molded at 5 MPa for 5 min. On the other hand, *V_f_* 48.7% of the molded products were formed at 10 MPa. The value of *V_f_* for each molded product formed at 5 MPa reflects the designed *V_f_* of 41.5% for the spread commingled yarn fabric shown in Table 2, and is almost the same, regardless of the forming time. On the contrary, the value of *V_f_* for the molded product formed at a molding pressure of 10 MPa was 48.7%, which was much higher than the designed *V_f_* value. This is probably because there is more leakage of matrix resin during molding at the higher molding pressure of 10 MPa.

### 3.2. Tensile Test

The stress-strain (S-S) curves obtained from the tensile test is shown in Figure 7. Each sample shows a test result close to the average of the five tests. It can be seen that the S-S curves of the product molded at 5 MPa is linear up to the maximum tensile stress regardless of the molding time. On the other hand, the S-S curve of the product molded at 10 MPa is lower. Figure 8 shows the relationships between tensile strength, tensile strain, tensile modulus, and molding time. The tensile strength and breaking strain slightly decreased with the increased molding time, but the tensile modulus did not change. The longer the molding time, the lower the elongation at break and, consequently, the lower the breaking stress. This may be due to thermal degradation of the nylon matrix. Figure 9 shows the results of varying the molding pressure while the molding time was fixed at 30 min. As shown in Table 7, the higher the molding pressure, the more the nylon matrix resin is flowed out of the molded product. The volume fraction of fibers also increases, which is thought to increase the fracture strength and elongation at break. However, the reason for the reduction in the modulus at the molding pressure of 10 MPa is probably due to the disorderly orientation of the carbon fiber in the tensile direction. In the case of spread yarn fabric, it is most important to prevent the flow of resin from causing the carbon fibers to be distorted.

There have been few reports about the mechanical properties of composites made from commingled carbon/nylon fiber fabrics, Toyota et al. evaluated the mechanical properties of fabrics made from spread commingled carbon fiber yarns coated with nylon resin, and reported that the fabrics made from this had a 20% higher elastic modulus and strength than those made from unspread yarns [27]. The tensile modulus of carbon fiber TR50S12L is 235 GPa, therefore, if the volume fraction is 39 % and the nylon fibers’ modulus are ignored, the tensile modulus of composites will be 91.7 GPa if the carbon fibers are aligned in the tensile direction. On the other hand, in plain weave, half of the carbon fibers are at 90° to the tensile direction, which means that modulus of the 90° fibers is almost zero. Therefore, the modulus is further halved to 45.8 GPa. The tensile modulus in Figure 8c ranged from 40 to 45 GPa, which was almost exactly as expected.

### 3.3. Four-Point Flexural Test

Figure 10 shows the S-S curves obtained from the four-point flexural test. Each sample shows the average results of the six samples tested. It can be seen that the S-S curves are linear up to the maximum breaking flexural stress under any of the molding conditions. Figure 11 shows the relationship between flexural strength, flexural modulus, flexural strain at break, and molding time. The results showed that the molding time had no effect on the flexural strength, flexural modulus or flexural strain at break. Kawabe et al. applied a flexural test to a molded body with a thickness of 1 mm, consisting of 19 layers, each made of alternating 12 K carbon fiber unidirectional sheets spread to a width of 16 mm and nylon 6 film with a thickness of 25 μm. As a result, the flexural strength increased as the molding time increased, but the increase after 5 min was slight. The tensile properties in Figure 6 decreased with increasing molding time, but the flexural properties in Figure 9 were not affected by molding time. Considering that the impregnation of the nylon film on the spread 12 K carbon fiber bundle is almost complete in 5 min [26], the resin impregnation in the commingled yarn, consisting of nylon and 12 K carbon fiber in this study, was sufficiently complete in less than 5 min. In the flexural test, we did not observe any decrease in strength due to nylon resin degradation over the long molding time observed in the tensile test. This is presumably because the spread woven fabric was used. All fractures in the flexural tests were on the tensile side, and the composite failed without interlaminar shear failure or buckling failure. This may be because the strength retention effect of the weft yarn due to the woven structure is greater than the degradation of the resin due to the long molding time, and this effect was not observed within the error range. With a higher molding pressure, the flexural properties become slightly better, but these were within the margin of error (Figure 12).

## 4. Discussion

Figure 13 and Figure 14 show cross-sectional images of the products molded at 5 MPa and 10 MPa, respectively. The white horizontal stretch in each image shows the weft yarn of the carbon fiber, the white dot shows the warp yarn of the carbon fiber, and the gray area shows the matrix resin, Nylon-MXD10. The matrix resin was sufficiently impregnated into the carbon fiber bundle, and no voids were observed. Figure 15 shows the effects of molding time and pressure on the thickness of the commingled yarn after molding. Even with a molding time of 5 min, the nylon fibers are completely melted, whereas the carbon fiber bundle is still somewhat swollen. Therefore, 10 min is the optimum molding time, even for the spread fabric.

The impregnation rates of resin into the carbon fiber of the carbon fabric/film laminate, shown in Figure 16a, and the commingled yarn fabric shown in Figure 16b, were investigated. Ueda et al. used the Darcy rule to evaluate the impregnation rate of resin on carbon fiber [3]. Using Ueda et al.’s formula in Equation (1) and calculating the distance between the commingled nylon and carbon fibers *l*_0_ as 10 μm, we found that the resin was completely impregnated into carbon fibers in 20 s at a molding pressure of 1 MPa preheating (Figure 17). This is example one of an equation:(1)$$I={\left(\frac{2k(P_{m}-P_{0}}{\eta l_{0}^{2}}\right)}^{\frac{1}{2}}\times t^{\frac{1}{2}}$$

*t*: molding time, *η*: resin viscosity (120 Pa·s), *l*_0_: distance from the resin flow front to the center of the fiber bundle (10 µm commingled yarn, 100 µm film), *P_m_*: press pressure (1~10 MPa), *P_0_*: atmospheric pressure (0.101 MPa), *k*: constant (1.83 × 10^−14^). On the other hand, if distance between carbon fiber and nylon 6 resin or fiber; *l*_0_ is set to 100 μm, assuming the lamination of the spread carbon fiber and resin film, it took 3 min and 20 s for the resin to completely impregnate the carbon fiber under a molding pressure of 10 MPa (Figure 17).

Motochika et al. studied the resin impregnation rate of the commingled carbon fiber yarn and nylon fiber yarn used in this study. They found that the resin impregnation rate after 1 min was 98.5% (not impregnated rate: 1.5%) [29]. It can be deduced that formation of the woven fabric made of spread commingled yarns can be completed in a very short time.

In terms of the impregnation rate of the resin in the carbon fiber, the rate is assumed to be the same whether the fabric of commingled yarns is manufactured from spread yarns or not. On the other hand, as shown in Figure 16, in the case of film stacking procedure (Figure 16a), which is not a commingled yarn but a carbon fiber bundle with resin layers above and below, it is useful to reduce the thickness of the fiber bundle. The fiber bundle was spread in order to increase the rate of thermoplastic resin penetration into the carbon fiber. The woven sheet in Figure 16a has great rigidity because the resin is in the shape of film. However, fabrics made of commingled yarns are flexible. In addition, the flexibility is more pronounced for spread fabrics. The woven fabric made by spread carbon fiber bundles has the merit of maximizing the original strength and modulus of carbon fibers because it can reduce the curvature of fibers at the intersection of warp and weft yarns.

Takahashi et al. compared the transverse tensile strengths of carbon fiber and PEEK commingled yarn fabrics, carbon fiber yarn and PEEK yarn fabrics, and even carbon fiber impregnated PEEK prepreg [30]. The results showed that the order of tensile modulus was CF/PEEK prepreg > CF/epoxy prepreg > CF/PEEK commingle > CF/PEEK fabric > PEEK resin. The breaking stress was CF/PEEK prepreg > CF/PEEK commingle > CF/epoxy prepreg > PEEK resin > CF/PEEK fabric.

In the present study, CF/Nylon commingled yarn was used and woven with spread fiber. In the future, it will be necessary to develop a loom suitable for fiber spreading and improve the manufacturing method of commingled yarn so that the results can be compared with fabrics of commingled yarn without fiber spreading, the effect of carbon fiber sagging caused by fiber spreading, and the optimum width of fiber spreading.

## 5. Conclusions

We investigated whether woven fabric consisting of spread commingled yarns made of carbon and nylon fibers could shorten the impregnation distance from matrix resin to carbon fibers and thus shorten the molding time of CFRTP. Woven fabrics made of commingled nylon fiber and carbon fiber yarns are easy to handle, and minimize the decrease in the elastic modulus of carbon fiber due to crimping, which is characteristic of woven fabrics. There are few reports on the design of woven fabrics by spread carbon fiber bundles consisting of commingled yarns.
The impregnation speed of the nylon resin into the carbon fiber bundles were very fast: less than 1 min. As the molding time increased, the tensile strength and tensile fracture strain slightly decreased, indicating the deterioration of the the nylon resin. As the molding pressure increased, the amount of matrix resin flowing out increased and the fiber volume fraction of the molded product increased, resulting in an increase in tensile strength and tensile strain at break.The effects of molding time on flexural strength, flexural modulus, and flexural fracture strain were negligible.As a result of the cross-sectional observation conducted to confirm the impregnation state of the matrix resin, no voids were observed in the molded products regardless of the molding time or molding pressure, indicating that resin impregnation into the carbon fiber bundles of the spread commingled yarn fabric was completed at a molding pressure of 5 MPa and a molding time of 5 min.The combination of commingled yarn, spread fiber, and woven fabric suggested the possibility of flexible and easy-to-handle thermoplastic CFRP prepreg.This is the first study on CFRTP using spread commingled carbon fiber and nylon fiber yarns as woven fabric.

## Figures and Tables

**Figure 1 polymers-13-03206-f001:**
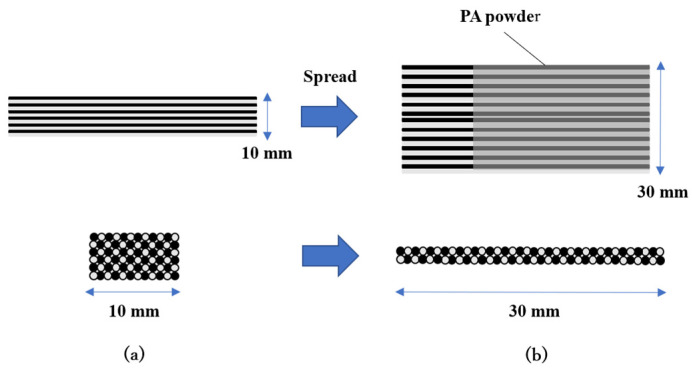
Image of spread of commingled yarn. (**a**) Standard commingled yarn; (**b**) Spread commingled yarn.

**Figure 2 polymers-13-03206-f002:**
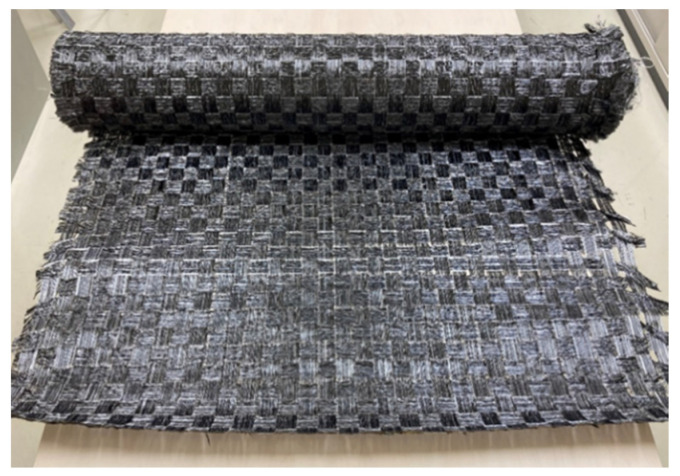
Woven fabric using spread commingled yarn.

**Figure 3 polymers-13-03206-f003:**
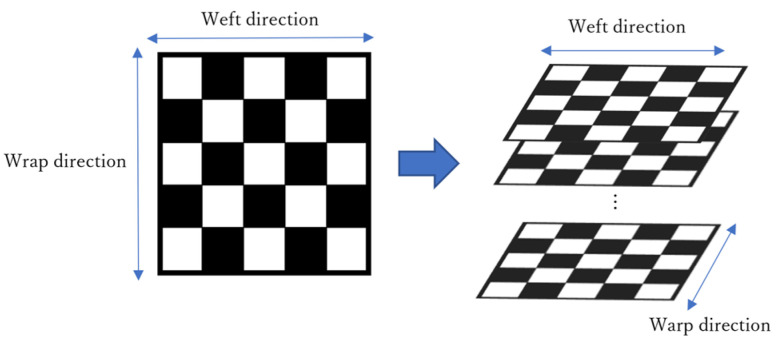
Laminated configuration of woven fabric using spreading.

**Figure 4 polymers-13-03206-f004:**
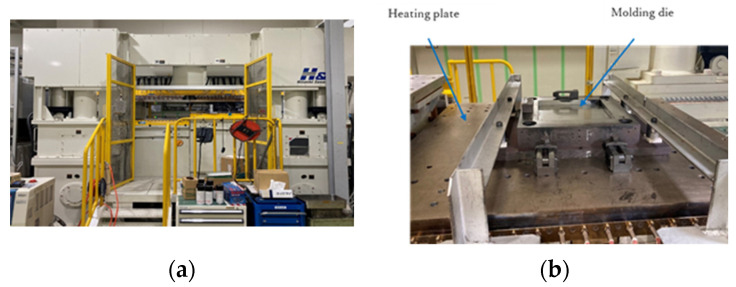
(**a**) Press-molding machine; (**b**) Press-molding die attached to heating plate.

**Figure 5 polymers-13-03206-f005:**
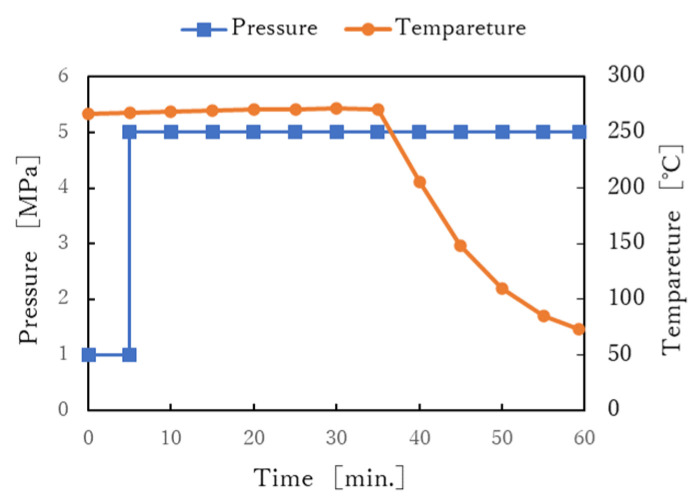
Profile of molding pressure and molding temperature.

**Figure 6 polymers-13-03206-f006:**
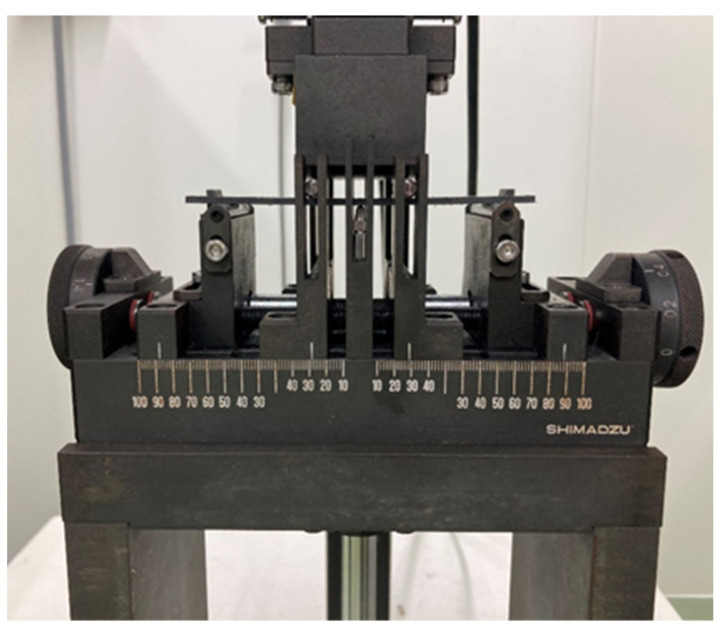
Four-point flexural test’s jig (Shimadzu Co. Ltd., Kyoto, Japan).

**Figure 7 polymers-13-03206-f007:**
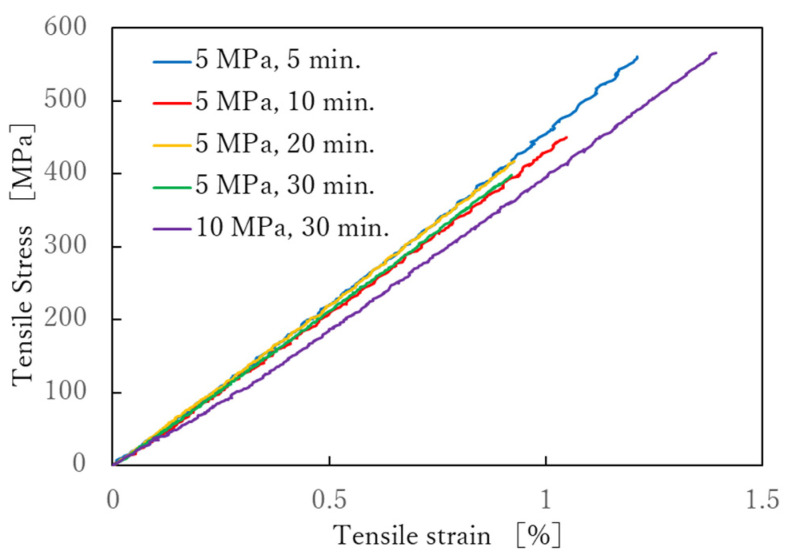
Stress-strain curves of the tensile test.

**Figure 8 polymers-13-03206-f008:**
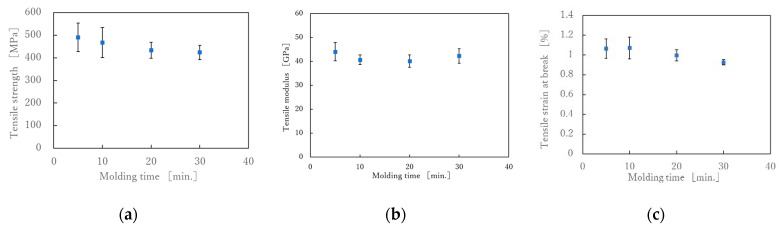
(**a**) Effect of molding time on tensile strength under molding pressure of 5 MPa; (**b**) Effect of molding time on tensile modulus under molding pressure of 5 MPa and (**c**) Effect of molding time on tensile strain at break under molding pressure of 5 MPa.

**Figure 9 polymers-13-03206-f009:**
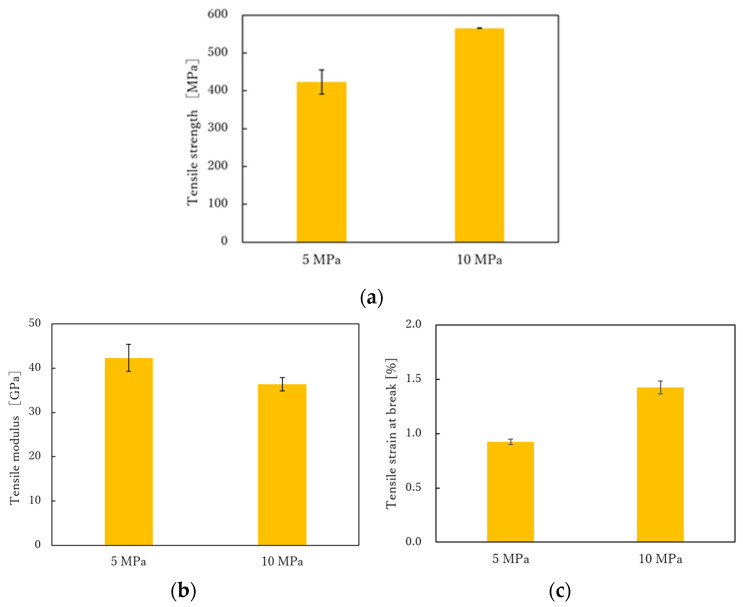
(**a**) Effect of molding pressure on tensile strength under molding time of 30 min; (**b**) Effect of molding pressure on tensile modulus under molding time of 30 min and (**c**) Effect of molding pressure on tensile strain at break under molding time of 30 min.

**Figure 10 polymers-13-03206-f010:**
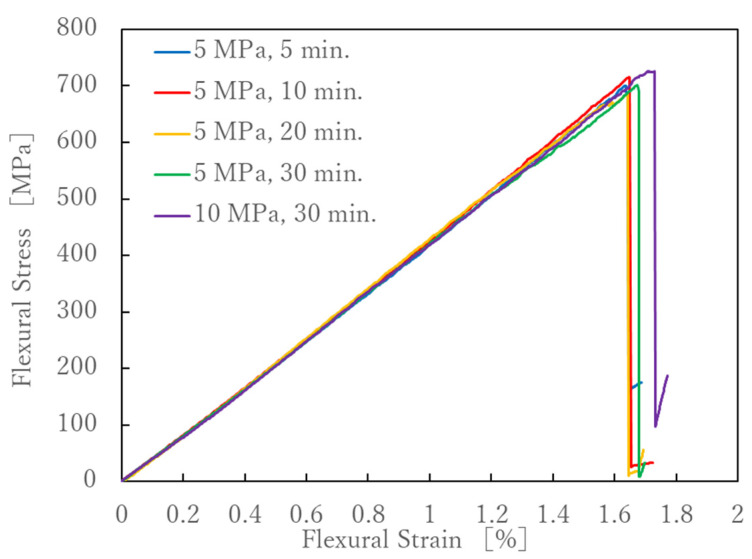
Stress-strain curves of the four-point flexural test.

**Figure 11 polymers-13-03206-f011:**
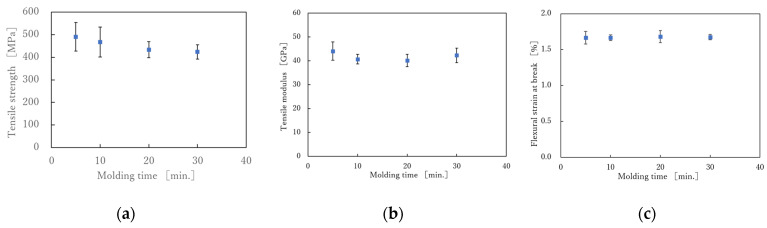
(**a**) Effect of molding time on flexural strength under molding pressure of 5 MPa.; (**b**) Effect of molding time on flexural modulus under molding pressure of 5 MPa and (**c**) Effect of molding time on flexural strain at break under molding pressure of 5 MPa.

**Figure 12 polymers-13-03206-f012:**
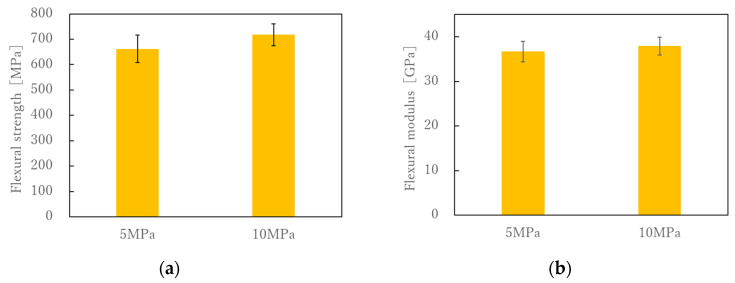
(**a**) Effect of molding pressure on flexural strength under molding time of 30 min; (**b**) Effect of molding pressure on flexural modulus under molding time of 30 min.

**Figure 13 polymers-13-03206-f013:**
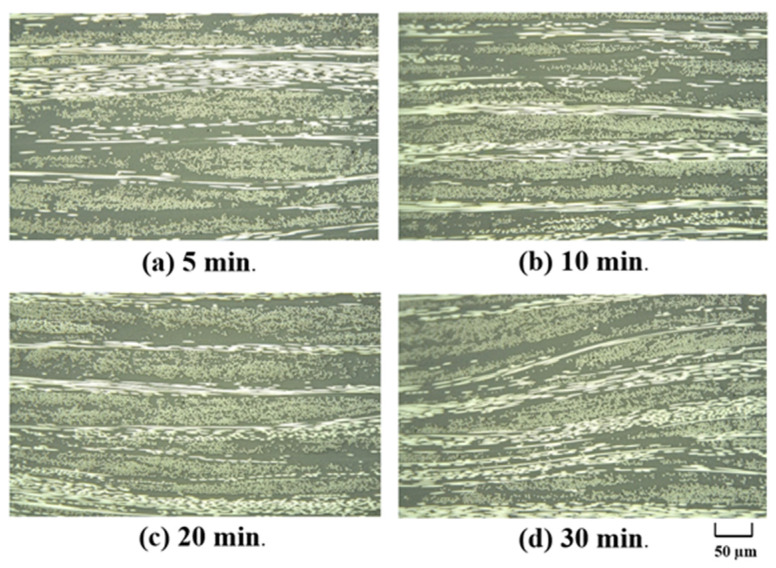
Cross-sectional photographs of composite materials made at different molding times under molding pressure is 5 MPa.

**Figure 14 polymers-13-03206-f014:**
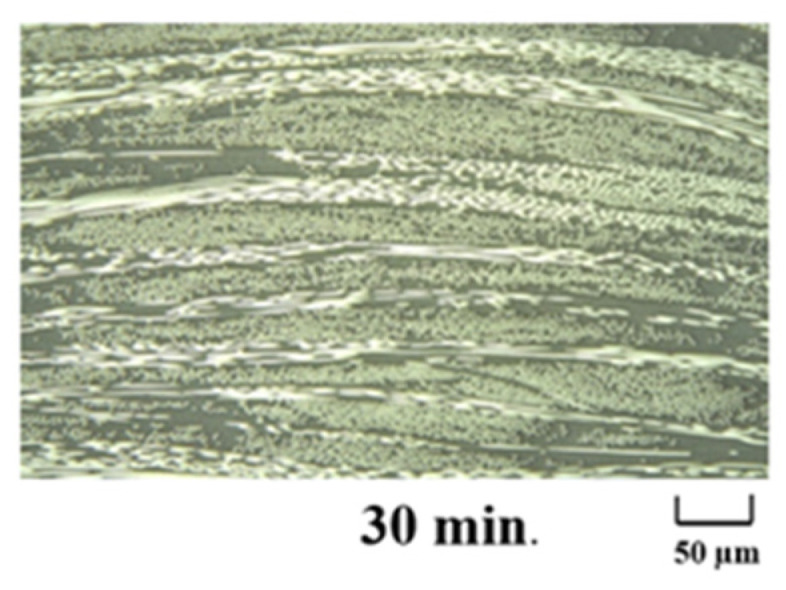
Cross-sectional photographs of composite materials made at molding time is 30 min under molding pressure is 10 MPa.

**Figure 15 polymers-13-03206-f015:**
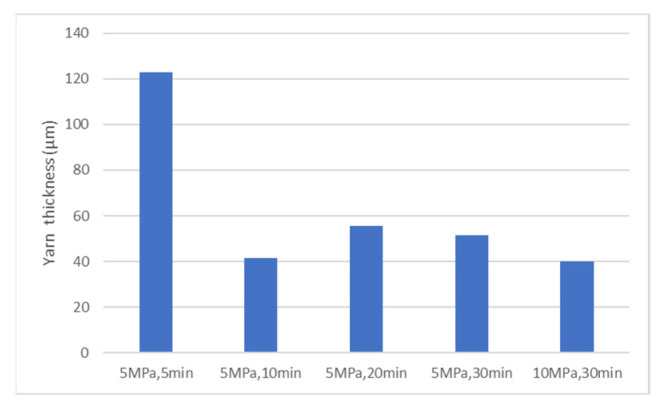
Effects of molding time and pressure on the thickness of the commingled yarn after molding.

**Figure 16 polymers-13-03206-f016:**
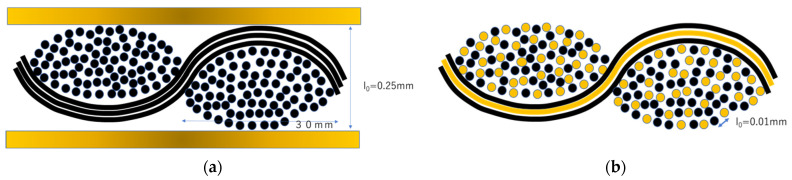
(**a**)Molding of composite materials from carbon fiber fabric and thermoplastic film; (**b**) Molding of composite materials from commingled yarns.

**Figure 17 polymers-13-03206-f017:**
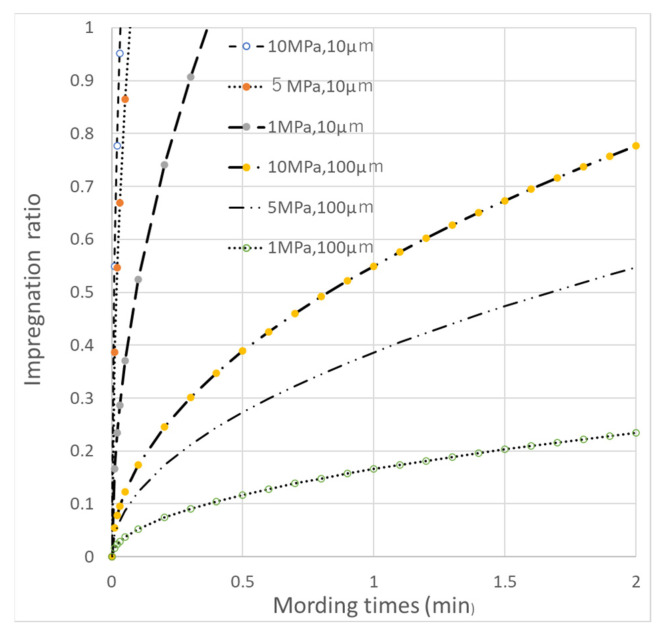
Impregnation ratio as a function of molding pressure and *l_0_*.

**Table 1 polymers-13-03206-t001:** Specifications of commingled yarn (Manufacturer; Kajirene Co. Ltd., Kanazawa, Japan).

Matrix resin	Low-water-absorbing nylon MXD10 (LEXTER manufactured by Mitsubishi Gas Chemical Co. Ltd., Japan) 1532 filaments (3.125 d mono-filament)
Carbon fiber	PAN CF (Mitsubishi Chemical Co. Ltd. TR50S12L, Japan) 12,000 filaments
Designed value of fiber volume content ($V_{f}$)	50%

**Table 2 polymers-13-03206-t002:** Specifications of woven fabric using commingled yarn (Manufacture; Harmoni Industry Co. Ltd., Fukui, Japan).

Weave Structure	Plain
basis weight	107.5 g/m^2^
Thickness	244.6 µm
Fiber volume content ($V_{f}$)	41.5 %

**Table 3 polymers-13-03206-t003:** Specifications of laminated materials.

Length	Width	Layer Number
245 mm	245 mm	30 ply

**Table 4 polymers-13-03206-t004:** Press molding conditions.

Molding Temperature	Preheating	Molding Time	Molding Pressure
270 °C	1 MPa, 5 min	5 min, 10 min 20 min, 30 min	5 MPa, 10 MPa (30 min)

**Table 5 polymers-13-03206-t005:** Conditions of tensile test.

Test Specimen Longitudinal Direction		Warp Direction
Test specimen dimensions	length	240 mm
	width	25 mm
	thickness	2 + 0.4 mm
Distance between two lines of sight		50 mm
Distance between tabs		150 mm
Tab Length		45 mm
Tensile test speed		2 mm/min.
Sample number		5

**Table 6 polymers-13-03206-t006:** Conditions of four-point flexural test.

Test Specimen Longitudinal Direction		Warp Direction
Test specimen dimensions	length	100 mm
	width	15 mm
	thickness	2 ± 0.4 mm
Distance between fulcrums		88 mm
Distance between indenters		29 mm
fulcrums radius		3 mm
indenters radius		3 mm
Flexural test speed		5 mm/min.
Sample number		6

**Table 7 polymers-13-03206-t007:** Fiber volume fraction of each molded products.

Molding Time	5 min	10 min	20 min	30 min	30 min
Molding pressure	5 MPa				10 MPa
Fiber volume fraction ($V_{f}$)	38.9%	38.7%	38.3%	38.6%	48.7%

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
