# Peer review of "Mechanical Properties of Thermoplastic Composites Made of Commingled Carbon Fiber/Nylon Fiber"

_polymers, 2021, doi:10.3390/polym13193206_

Round 1
Reviewer 1 Report
The authors have conducted a very systematic study wherein they have presented the results quite clearly with relevant scientific discussions wherever applicable. However, please find below a few suggestions/comments which may help in improving the overall quality of the manuscript.
- It is better to avoid abbreviations in an abstract, especially if they have not been defined. The case in point is, CFRTP, which is not defined. Please check.
- “In this study, we investigated whether an open weave fabric consisting of commingled
yarns made of carbon and nylon fibers could shorten the impregnation distance of resin to carbon fibers, and there are few reports on the design of fabrics by opening carbon fiber bundles consisting of commingled yarns.” I guess the above statement is confusing. Should be revised and rephrased, especially the second part. - I guess, lines 40 to 42 and lines 48 to 50 are repeated. Please revise.
- CFRTP is still not defined in the Introduction of the manuscript as well. Please check.
- What is PAN?
- Line 85 and 86, “Opening” should be “Opened”
- Section 2.1 should be revised in terms of the tenses used.
- “A heating and cooling press molding machine (Hitachi Zosen Fukui Corporation 110 UFHS2500) was used as a press molding machine.”….Please revise.
- I am fit confused by the usage of the word “Opening” throughout the manuscript. Please clarify.
- Is there any basis on the selection of the parameters such as the molding temp (270) and the molding pressures?
- The quality of the legends and labels should be improved in Figures 8, 9, 10, 11 and 12 for clarity.
- No specific reason has been given for the reduction in the Tensile modulus at 10MPa, inspite of an increase in the tensile strength and tensile strain.
- It will be good if the authors can compare the properties obtained in their current study with the available studies in the literature to show the significance of the results. They may think of presenting it in the form of a summary table.
Author Response
Dear Peer Reviewer
I am very grateful for your pertinent comments.
- It is better to avoid abbreviations in an abstract, especially if they have not been defined. The case in point is, CFRTP, which is not defined. Please check.
The following is a description of the process.
Line 13 and line 37
carbon fiber reinforced thermoplastic (CFRTP)
2)“In this study, we investigated whether an open weave fabric consisting of commingled
yarns made of carbon and nylon fibers could shorten the impregnation distance of resin to carbon fibers, and there are few reports on the design of fabrics by opening carbon fiber bundles consisting of commingled yarns.” I guess the above statement is confusing. Should be revised and rephrased, especially the second part.
The following description has been changed.
In this study, we examined whether spreaded commingled carbon fiber/nylon fiber yarns could reduce the impregnation distance or not, as there are few reports on this.
3)I guess, lines 40 to 42 and lines 48 to 50 are repeated. Please revise.
Duplicates ( Lines 40 to 42) have been removed. 
“Therefore, if it is possible to give thermoplastic CFRP excellent properties such as high strength and a high elastic modulus comparable to those of thermoset CFRP, it will be expected to be a material with high productivity and excellent recyclability.”
4)CFRTP is still not defined in the Introduction of the manuscript as well. Please check.
The following is a description of the process.
“carbon fiber reinforced thermoplastic (CFRTP)”
5)What is PAN?
Additional explanation was added to Line 81.
polyacrylonitrile (PAN)
6)Line 85 and 86, “Opening” should be “Opened”
I've changed the wording to "spread" or "spreaded" because "opening" and "opened" are misleading.
7)Section 2.1 should be revised in terms of the tenses used.
“A heating and cooling press molding machine (Hitachi Zosen Fukui Corporation 110 UFHS2500) was used as a press molding machine.”….Please revise.
Additional explanation was added to Line 109-115.
A heating and cooling press molding machine (Hitachi Zosen Fukui Corporation UFHS2500) was used as a press molding machine. Figure 4 shows a photograph of the machine. The heating plate and the cooling plate were lined up next to each other, and a die lifter was installed in front of and behind the hot plate. As a result, heated molds were transported on a cooling plate using a mold lift for quick cooling. For the molding, a male−female mating type was used. The mold area dimensions were 250 mm square.
8)I am fit confused by the usage of the word “Opening” throughout the manuscript. Please clarify.
I've changed the wording to "spread" or "spreaded" because "opening" and "opened" are misleading.
9)Is there any basis on the selection of the parameters such as the molding temp (270) and the molding pressures?
The citation literature that determined the molding conditions has been added to line 119.
The melting point of the nylon MDX used here is 230 to 240℃. Molding is performed at temperatures below 300℃ to prevent polymer degradation. Reference is made to literature [23], which uses a molding temperature of 270℃ for nylon 6. Molding pressure varies depending on the molding machine mold, the number of prepreg sheets stacked, and the melting viscosity of the resin. Normally, 1 to 20 MPa is used. This is also based on the same literature. I have included the literature in the text.
10)The quality of the legends and labels should be improved in Figures 8, 9, 10, 11 and 12 for clarity.
The font in the figure has been enlarged to make it easier to see.
11)No specific reason has been given for the reduction in the Tensile modulus at 10MPa, in spite of an increase in the tensile strength and tensile strain.
Additional explanation was added to Line 194-198.
“But, the reason for the reduction of the modulus at the molding pressure of 10MPa is probably due to the disorderly orientation of the carbon fiber in the tensile direction. In the case of spreaded yarn fabric, it is most important to prevent the flow of resin from causing the carbon fibers to be distorted.”
12)It will be good if the authors can compare the properties obtained in their current study with the available studies in the literature to show the significance of the results. They may think of presenting it in the form of a summary table.
Additional explanation was added to Line 199-209.
“There have been few reports about the mechanical properties of composites made from commingled carbon/nylon fiber fabrics, Toyota et al. [27] evaluated the mechanical properties of fabrics made from spreaded commingled carbon fiber yarns coated with nylon resin, and reported that the fabrics made from it had 20% higher elastic modulus and strength than those made from unspreaded yarns. The modulus of elasticity of carbon fiber TR50S12L is 235 GPa, therefore, the volume fraction is 39% and nylon fibers modulus are ignored, the composite modulus will be 91.7 GPa, if the carbon fibers are aligned in the tensile direction. On the other hand, in plain weave, half of the carbon fibers are at 90° to the tensile direction, which means that modulus of 90°fibers are almost zero, so the modulus is further halved to 45.8 GPa. The tensile modulus in Figure 8(c) was 40 to 45 GPa, which was almost exactly as expected.”

Reviewer 2 Report
Adequate topic and a good set of experimental data. The authors are still required to mention the new aspects of the research and refer to very recent literature
References can include recent references
English requires improvement (examples are given in square brackets [[ ]])
Introduce the full term then acronyms CFRTP
Fiber-opening treatment of commingled yarns consisting of thermoplastic nylon fibers 9 and carbon fibers could produce superior CFRTP,
The impregnation speed of the nylon 14 resin on the carbon fiber was very fast, less than 1 minute.
2.1. Material [[Materials]]
Figure 9 caption…a)……b)…..[[ and ]]] c)
Same for the other Figures
Figure 12. (a) Effect of molding pressure on flexural strength under molding time of 30minutes; (b) Effect of molding 237 pressure on flexural modulus under molding time of 30 minutes.
Figure 12. Effect of molding pressure on (a) flexural strength and (b) flexural modulus [[ under ]] molding time of 30minutes.
Figure 13. Cross-sectional photographs of composite materials made [[ at ]] different molding times [[[ at ]] a molding pressure of 5 MPa.
Author Response
Dear Peer Reviewer
Thank you very much for your kind comments.
We have made the following revisions and hope you will consider them.
1) The authors are still required to mention the new aspects of the research and refer to very recent literature
We added the latest literature. (References 10-12, 20-23) and we added the novelty of this study by comparing it with the latest papers to Line 338-341.
2) English requires improvement (examples are given in square brackets [[ ]])
We have corrected the part in English that you pointed out.
3) Introduce the full term then acronyms CFRTP
The following is a description of the process.
carbon fiber reinforced thermoplastic (CFRTP)
